# Seasonal and Spatial Dynamics of Fungal Diversity and Communities in the Intertidal Zones of Qingdao, China

**DOI:** 10.3390/jof9101015

**Published:** 2023-10-14

**Authors:** Wei Li, Qi Li, Zhihui Pan, Gaëtan Burgaud, Hehe Ma, Yao Zheng, Mengmeng Wang, Lei Cai

**Affiliations:** 1College of Science, Shantou University, Shantou 515063, China; cheryli@stu.edu.cn (Q.L.); wangmmouc@163.com (M.W.); 2Guangdong Provincial Key Laboratory of Marine Biotechnology, Shantou University, Shantou 515063, China; 3College of Marine Life Sciences, Ocean University of China, Qingdao 266100, China; zhihuipann@163.com (Z.P.); maheouc@163.com (H.M.); zhengyao1ouc@163.com (Y.Z.); 4Laboratoire Universitaire de Biodiversité et Écologie Microbienne, Frech National Research Institute for Agriculture, Food and Environment (INRAE), Université de Bretagne Occidentale, F-29280 Plouzané, France; gaetan.burgaud@univ-brest.fr; 5State Key Laboratory of Mycology, Institute of Microbiology, Chinese Academy of Sciences, Beijing 100101, China

**Keywords:** coastal region, fungal richness, community structure, trophic mode, spatiotemporal pattern

## Abstract

Intertidal zones comprise diverse habitats and directly suffer from the influences of human activities. Nevertheless, the seasonal fluctuations in fungal diversity and community structure in these areas are not well comprehended. To address this gap, samples of seawater and sediment were collected seasonally from the estuary and swimming beaches of Qingdao’s intertidal areas in China and were analyzed using a metabarcoding approach targeting ITS2 rDNA regions. Compared to the seawater community dominated by *Ciliophora* and Agaricomycetes, the sediment community was rather dominated by Dothideomycetes and Eurotiomycetes. Furthermore, the seawater community shifted with the seasons but not with the locations, while the sediment community shifted seasonally and spatially, with a specific trend showing that *Cladosporium*, *Alternaria*, and *Aureobasidium* occurred predominantly in the estuarine habitats during winter and in the beach habitats during spring. These spatiotemporal shifts in fungal communities’ composition were supported by the PERMANOVA test and could be explained partially by the environmental variables checked, including temperature, salinity, and total organic carbon. Unexpectedly, the lowest fungal richness was observed in the summer sediments from two swimming beaches which were attracting a high influx of tourists during summer, leading to a significant anthropogenic influence. Predicted trophic modes of fungal taxa exhibited a seasonal pattern with an abundance of saprotrophic fungi in the summer sediments, positively correlating to the temperature, while the taxa affiliated with symbiotroph and pathotroph-saprotroph occurred abundantly in the winter and spring sediments, respectively. Our results demonstrate the space–time shifts in terms of the fungal community, as well as the trophic modes in the intertidal region, providing in-depth insights into the potential influence of environmental factors and human activity on intertidal mycobiomes.

## 1. Introduction

Marine intertidal areas are impacted by a wide range of physiochemical and biological forcings providing a non-trivial effect on ecological processes, e.g., biogeochemical cycling of carbon, nitrogen (N), phosphorus (P), sulfur, and iron [1]. Estuarine and coastal intertidal areas are vital in influencing global ocean processes, emphasizing the ecological significance of these habitats. However, these areas are experiencing growing human populations and exploitation pressures, resulting in a range of significant anthropogenic threats [2,3]. Substantial habitats including mangroves, sand beaches, estuarine, shallow coral, and salt marsh are well represented in intertidal zones, hosting a rich microbial biodiversity and high endemicity [3,4]. Recently, a review listed 1898 marine fungal taxa recorded from a wide diversity of habitats (e.g., mangrove, salt marsh, and deep-sea) and substrates (e.g., drifting wood, sediment, plants, animals, and algae) and occurring as saprobes, symbionts, parasites, or mutualists [5]. Another recent review focusing on an intertidal habitat, i.e., salt marsh, has reported about 500 fungal taxa associated with salt marsh halophytes worldwide [6], indicating the significant weight of this coastal dynamic ecosystem in terms of fungal resources. Despite this, the fungal diversity in such intertidal zones is still largely underestimated due to the congenital limitation of the culture-dependent approach allowing to generate and describe the listed marine fungal taxa, as the cultivation efficiency of living microbial cells using standard methods is estimated to be between 0.001 and 1% [7,8].

Limited molecular surveys using next-generation sequencing (NGS) approaches have shown that intertidal fungi are considerably more diverse than previously thought in terms of community composition and ecological roles. Using 454 pyrosequencing-targeted ITS1 and ITS2 rDNA regions, Arfi et al. [9] reported that Agaricomycetes (Basidiomycota) was the dominant fungal group in the anoxic-sulfidic sediments collected from the mangroves of Saint Vincent Bay (New Caledonia), where fungi were prone to degrade organic matter. A complementary study using multiple gene markers revealed a clear host-specificity of a fungal community among mangrove trees and a differentiation in community composition from mangrove sediments (mainly comprised of saprophytic fungi) to healthy leaves (pathogenic fungi) [10]. Using another genetic marker, i.e., LSU rRNA, Picard [11] provided a different representative of the fungal communities in the intertidal wetlands of North Carolina, with an appreciably large part of chytrids that are known to parasitize phytoplankton in various marine habitats, with a global tropism towards cold waters [12,13,14]. This suggests a potential parasite–host relationship that is likely to play an important role in shaping intertidal ecosystems. A different picture was observed in Chinese intertidal sediments, where Ascomycota and Basidiomycota were the dominant phyla from a fungal ITS2 rDNA dataset [15] and suggested to mainly serve as parasites or symbionts of marine flora and fauna, as well as decomposers of organic materials, as predicted by functional guilds [16].

Time-series assessment of coastal mycoplankton communities is essential in boosting the knowledge of their biology and ecology [17,18]. Several NGS-based studies have shown that mycoplankton diversity and communities in coastal waters display seasonal dynamics as a response to biological and abiotic factors [18,19,20]. For example, Taylor and Cunliffe [18] observed a seasonal synchronization between chytrid and diatom blooms in the coastal waters off Plymouth (UK), seemingly highlighting parasitism between marine chytrids and diatoms. Recently, Wang et al. [20] revealed seasonal shifts in fungal diversity and community composition in the water column of the Yellow Sea, as driven by the combined effect of hydrographic conditions and riverine inputs. However, NGS-based studies focused on the seasonality of intertidal fungal communities are rare, despite an atypical seasonality of the fungal communities in the intertidal flats of North Carolina, as represented by more diverse fungal communities in cooler months [11]. Thus, the extremely limited information hinders our understanding of seasonal dynamics of intertidal mycobiomes and the influence of environmental variables.

As a coastal city in the Shandong Peninsula, Qingdao directly faces the Yellow Sea in the south and east, with a clear seasonality in climate. Regarding this, we hypothesized a seasonal differentiation in the fungal communities in tidal flats, as previously documented in coastal waters [18,19,20]. Furthermore, this city is famous for its tourist spots, popular beaches, and scenic seaside, which attract a huge influx of tourists; for example, it received about 109 million tourists in 2019 [21]. Recent studies have found that several bathing beaches in Qingdao have been seriously contaminated by marine litter and microplastics due to beachgoers and recreational activities, especially during summer, the peak tourist season [21,22]. Also, the urban development and population expansion of Qingdao produce more and more domestic and industrial wastewaters, which have influenced the coastal environments, especially in Jiaozhou Bay, where human activities have caused a significant increase of N and P in waters [2,23] that can directly influence mycoplanktonic diversity [17]. Here, we selected two estuarine habitats in the Jiaozhou Bay, i.e., the Baisha River (EBS) and the Dagu River (EDG), and two sand beaches, i.e., the First swimming beach (FSB) and the Shilaoren swimming beach (SSB), as our targeted intertidal zones to explore the spatiotemporal dynamics in fungal diversity, community composition, and potential functional roles, as well as the impacts of environmental factors and human activities, using an ITS2 metabarcoding approach.

## 2. Materials and Methods

### 2.1. Sample Collection

Along the coastline of Qingdao, four locations, i.e., FSB, SSB, EBS, and EDG, were selected to collect water and sediment samples (Figure 1). Both FSB and SSB are sand beaches that strongly suffer from human activities due to the massive influx of tourists during summer [21,22]. Available data show that summer is the peak tourist season, and the total number of tourists during July and August accounted for 42.4% of the annual tourist volume in 2019, which partially explained the fact that the abundance of microplastics in sediments during the summer was higher compared to the winter [21]. The two estuarine locations, i.e., EBS and EDG, which are located in the Jiaozhou Bay, are muddy beaches and are highly influenced by riverine inputs of the Baisha and Dagu rivers, respectively. Sediment sampling was conducted in the meso-tidal beach of the intertidal region, meaning that the sampling zones could be exposed to air at low tide and submerged in seawater at high tide [24]. An approximately rectangular (bathing beach) or irregular (estuarine habitat) area of about 200 m by 3 m–10 m at each location was designed as the sampling zone. In each sampling zone, a square area of about 1 m by 1 m was selected randomly as the sampling point, where five subsamples were collected and mixed thoroughly as a single sample. Surface sediments were removed, and the sediments at depths from 2 cm to 10 cm were collected with sterile sampling bags and carried to the laboratory, then stored at −80 °C for DNA analyses. Intertidal water sampling was conducted using sterile glass bottles, and a Sterivex filter (0.45-µm pore-size, Millipore, Billerica, MA, USA) was used to filter 1 L of water. The filters were immediately sealed and brought to the laboratory and were frozen at −80 °C.

Sampling was performed during the typical months of different seasons, i.e., December (winter) of 2020, March (spring), June (summer), and September (autumn) of 2021. We obtained 95 samples which comprised 80 sediment samples and 15 water samples. An elemental analyzer (EA3000, Euro Vector SpA, Milan, Italy) was used to measure the total organic carbon (TOC), total nitrogen (TN), and total phosphorous (TP) in the sediments. The temperature, salinity, and pH of the water column were measured using hand-held meters (SMART SENSOR, Dongguan, China) (Appendix A).

### 2.2. DNA Extraction and High-Throughput Sequencing

Genomic DNA was extracted from the sediments (0.5 g) and from 1 L of seawater using Sterivex filters (Millipore, Billerica, MA, USA), based on an optimized protocol of Collins et al. [25], as previously described [26]. The primers ITS3/ITS4 (ITS3: 5′-GCATCGATGAAGAACGCAGC-3′; and ITS4: 5′-TCCTCCGCTTATTGATATGC-3′) [27] were used to amplify the ITS2 rDNA regions. Each 20 µL of PCR reaction consisted of 10 µL of HotStarTaqMaster Mix (Qiagen, Hilden, Germany), 500 nM of each primer, 10 ng of template DNA, and nuclease-free water. The PCR conditions were as follows: 95 °C for 15 min, followed by 30 cycles of denaturation at 95 °C for 30 s, annealing at 50 °C for 30 s, and extension at 72 °C for 60 s, and a final extension at 72 °C for 10 min. For each sample, the PCR reactions were run in triplicate, then pooled together. The quantity of the amplicon DNA was estimated by running 5 μL on a 2% agarose gel. No PCR products were displayed in our negative controls using the same PCR conditions and primers. A GeneJET™ Gel Extraction Kit (Thermo Scientific, Waltham, MA, USA) was used to purify the amplified DNA. The amplicon libraries were generated using an NEB Next^®^ UltraTM DNA Library Prep Kit for Illumina (NEB, Ipswich, MA, USA). The ITS2 amplicons were sequenced using the NovaSeq PE250 (Personalbio Corporation, Shanghai, China) platform (paired-end reads 2 × 250 bp)’s high-throughput sequencing. The fastq and barcode files were deposited in the National Center for Biotechnology Information Sequence Read Archive (SRA) as BioProject ID: PRJNA964474.

### 2.3. Bioinformatic Analysis

FLASH (version 1.2.11) [28] and QIIME (version 1.4.3) pipeline (options: q = 19, r = 3, and *p* = 0.75) [29] were used to merge, filter, and assign paired-end reads to samples based on their unique barcode, respectively. Then, the filtered ITS2 reads were further dealt with using MOTHUR (version v1.48.0) [30] with the following options: no DNA ambiguities and a minimum sequence length >250 bases. Chimeras were removed using USEARCH (version 11.0.667_i86osx32) [31] against the UCHIME reference dataset (version 7.2) [32,33]. The nontargeted reads with a minlength setting of 99 bases were removed using ITSx (version 1.0.11) [34]. Operational taxonomic units (OTUs) were clustered at the 97% similarity level using CD-HIT (version 4.6.1) [35]. A taxonomical annotation of OTUs’ representative sequences was performed against the UNITE + INSDc database (version 8.2) using MegaBLAST searches (version 2.8.0+) [36] with conservative settings (word_size = 7, penalty = −3, and reward = 1). The OTUs with an e-value < e^−20^ and over 70% of the sequence length were classified as pertaining to the fungal kingdom and then assigned to a fungal genus, family, order, or class at 95%, 90%, 85%, and 80% sequence identities [27], respectively. The OTUs with an e-value > e^−20^ or assigned as unidentified fungi in the above step were further identified using a Naïve Bayesian Classifier (NBC) [37,38] with the Warcup ITS training set and UNITEv8_sh_dynamic.fasta as references, respectively. The NBC-based taxonomic assignment was to be accepted when the node bootstrap support was above 60% at the taxonomical levels ranging from genus to phylum.

For predicting potential ecological functions of the fungal OTUs, several ecological categories (e.g., animal pathogens, plant pathogens, and wood saprotrophs) within three trophic modes (i.e., pathotroph, symbiotroph, and saprotroph) were taxonomically parsed using the FUNGuild tool (version 1.1) [16]. Only the fungal OTUs that could be assigned at the genus level, i.e., with ≥95% sequence similarity to a reference sequence, were retained to identify guilds with the rank of possible confidence.

### 2.4. Statistical Analyses

To calculate the relative abundance, the number of reads assigned to an individual taxon (e.g., OTU and genus) within each sample was divided by the number of total reads within individual samples. For a comparative analysis of alpha-diversity (OTU richness and Shannon index), a dataset with the OTUs having no less than two reads was rarefied according to the lowest number of reads (18,341 reads) from our samples using the vegan package of R (version 4.3.1) [39]. After Leven’s test for the homogeneity of variance, variations between different groups were explored using a one-way ANOVA or Welch’s ANOVA followed by the adjusted *p* values of the Bonferroni method or the *p* values of the Games-Howell test, respectively (Appendix A). The relationships between the environmental factors and the fungal alpha diversity were estimated with Pearson’s coefficients. To avoid marginal effects induced by rare OTUs [33], only the OTUs represented by no less than five reads were retained to conduct multivariate analyses. Based on the Bray–Curtis dissimilarity matrix that had been generated from the relative abundance data, we used principal coordinate analysis (PCoA) plots to visualize the ordinations of community composition. The adonis2 function in the vegan package was used to check the influence of season, location, and substrate type (i.e., sediment and seawater) on community composition [40]. Mantel tests were performed to test the correlations between community dissimilarities and physicochemical parameters of seawater and sediment.

## 3. Results

### 3.1. The Difference of Environmental Parameters across Habitats

In the estuary habitat, seawater salinity of the EBS (average ± SD:21.68 ± 1.01 psu) and EDG (23.75 ± 3.12 psu) exhibited lower concentrations compared to the beach habitats, i.e., FSB (30.27 ± 0.68 psu) and SSB (30.48 ± 0.62 psu), suggesting a strong influence of riverine inputs. Furthermore, the two estuary habitats displayed seasonal and spatial dynamics of nutrient concentrations (i.e., TOC, TN, and TP) in sediments (Appendix A). For example, the highest level of TOC was observed at the EBS (0.0103 ± 7.65 × 10^−5^) and EDG (0.0109 ± 8.40 × 10^−4^) in spring and autumn, respectively, and the lowest level of TOC was observed during winter for the two estuaries. In summer, the two estuaries showed higher concentrations of TN in sediments than in other seasons. However, both of the beach habitats, FSB and SSB, showed lower levels of nutrient concentration in sediments than that of the EBS and EDG without an obvious seasonal shift (Appendix A).

### 3.2. Taxonomic Composition and α-Diversity of Fungal Community

After the filtering and removal of singletons, 14,279 OTUs were clustered from 6,350,822 reads. To ensure inter-sample comparability, the filtered reads were randomly resampled to reduce the number of reads in each sample to the lowest number (18,341). After resampling, the OTUs (7922) affiliated with the fungal kingdom were retained for downstream analysis (Appendix A). These fungal OTUs span seven known phyla, twenty-one classes, fifty-five orders, and two hundred thirty-five genera (Appendix A). Ascomycota and Basidiomycota appeared predominant, accounting for 64.1% and 31.3% of total fungal OTUs, respectively. Chytridiomycota (0.86%), Mucoromycota (0.74%), Rozellomycota (0.19%), Mortierellomycota (0.04%), and Aphelidiomycota (0.01%) were also detected in smaller proportions.

Distinct community compositions were revealed between seawater and sediment. In the intertidal waters, an uncertain class affiliated to the subphylum Pezizomycotina and Agaricomycetes appeared abundant in terms of relative read abundance, as mainly represented by *Ciliophora* and *Amphinema*, respectively (Figure 2). In sediments, the class Dothideomycetes, which abundantly consisted of *Cladosporium*, *Alternaria*, and *Aureobasidium*, occurred predominantly at the estuary habitats, i.e., EBS and EDG, in winter, as well as at the beach habitats, i.e., FSB and SSB, in spring. A contrasting trend displayed by the Eurotiomycetes (mainly comprised of *Penicillium* and *Aspergillus*) that prevailed at the EBS and EDG in most seasons, except winter, was observed. Moreover, the members of the class Eurotiomycetes were more abundant at the FSB and SSB in summer and winter than in spring and autumn.

For the total samples, the fungal OTU richness in the seawaters (average ± SD: 343.6 ± 146.7) was significantly higher than that in the sediments (193.6 ± 107.0) (Figure 3a). For most individual locations, however, a significant difference in fungal richness was not detected between the seawater and the sediment (Figure 3b–e). The highest levels of average fungal richness in the seawaters and sediments were observed in summer and autumn, respectively (Figure 3f,g). For estuary habitats, the highest richness (247.3 ± 94.8) occurred in the summer sediments of EBS, which exhibited a higher fungal diversity than that in the EDG sediments in terms of OTU richness and Shannon index (Figure 3h,i and Appendix A). For the beach habitats, lower levels of OTU richness were revealed in both the FSB and SSB sediments in summer compared to autumn and winter (Figure 3j,k). Among the environmental factors tested, salinity appeared to be weakly correlated with the OTU richness in sediments (*p* = 0.049, r = 0.220), while TN negatively correlated with the Shannon index (*p* = 0.040, r = −0.230). No significant correlations were found between the fungal diversity and other parameters, i.e., TOC, TP, TOC/TN ratio, pH, and temperature.

### 3.3. Space–Time Variation in Community Structure and the Influencing Factors

The PCoA plot indicated that the community structure of the seawater was distinct from that of the sediments (Figure 4a), which was further supported by the PERMANOVA test where the substrate type (i.e., sediment and seawater) significantly explained 4.6% of the variation in community structure (*p* = 0.001). For the sediment communities, both the season and the location were the important variables shaping community structure, and they explained 11.4% (*p* = 0.001) and 8.4% (*p* = 0.001) of the variation, respectively, highlighting an obvious space–time turnover in community composition. However, only the season exhibited a significant influence on the seawater communities, with 29.8% of variation explanation (*p* = 0.012). Among the environmental parameters investigated, temperature (r = 0.096, *p* = 0.002), salinity (r = 0.096, *p* = 0.045), pH (r = 0.137, *p* = 0.014), and TOC (r = 0.076, *p* = 0.032) had significant effects on the community structure in the sediments, according to Mantel tests. The temperature also strongly impacted the community structure in the seawaters (r = 0.371, *p* = 0.001).

Similarity analysis (1 – Bray–Curtis index) showed two distinct groups clustered using the seawater and sediment samples, respectively (Figure 4b), supporting our PCoA analysis. For the sediment samples, the pairs from autumn (0.208 ± 0.160) or winter (0.249 ± 0.187) were more similar in their community composition than the pairs from spring (0.101 ± 0.144) and summer (0.067 ± 0.095) (Appendix A). A similar trend occurred in the two beach habitats, FSB and SSB (Figure 4c). Moreover, the pairs from different locations during the same season differed from each other in community similarity (Appendix A). These results indicate a temporal and spatial variation of community structure at both the global and local levels.

### 3.4. Trophic Modes of Fungal Community in Intertidal Zone

A total of 3527 OTUs with 95% sequence similarity matching a reference sequence were used to estimate their trophic modes. Of them, 1369 OTUs (38.8%) were functionally assigned into saprotroph (438), pathotroph (329), symbiotroph (57), and two or more guilds (545), respectively. Interestingly, the proportions of OTUs assigned successfully appeared to be higher in the sediments of the beach habitats (FSB: 46.1%; SSB: 41.5%) than in that of the estuary habitats (EBS: 30.4%; EDG: 25.4%) (Appendix A). This indicates that beach habitats might harbor more active taxa than estuary habitats. However, such a hypothesis should be taken with caution due to the lack of completeness of the FUNGuild database which might result in numerous undetected OTUs of estuary sediment using this tool.

The trophic modes of the fungal community exhibited an obvious pattern, shifting with the season and the location. For both the sediment and seawater, the proportion of the saprotrophic OTUs was higher in the summer compared to the other seasons (Figure 5a). A spatial shifting of saprotroph was detected (F = 3.856, *p* = 0.013), as represented by the higher occurrence at FSB (31.9 ± 30.6%) than at EDG (7.5 ± 14.0%) in the summer sediments (Figure 5b,c). Moreover, we detected a positive correlation between the temperature and saprotrophic behavior in the sediments (t = 2.085, df = 78, *p* = 0.040). Symbiotroph (F = 6.231, *p* = 0.001) and Pathotroph-Saprotroph (F = 4.180, *p* = 0.009) showed a temporal turnover, with the highest occurrence in the sediments in winter (5.8 ± 6.6%) and spring (6.7 ± 5.6%), respectively.

## 4. Discussion

### 4.1. Intertidal Fungi Exhibit Spatiotemporal Variation in Taxonomic Composition

Our study revealed a significant proportion of fungal OTUs (~95%) belonging to the Ascomycota and Basidiomycota in the intertidal waters and sediments, mainly comprising Dothideomycetes, Eurotiomycetes, Exobasidiomycetes, Sordariomycetes, and Agaricomycetes. This is consistent with our previous study that captured fungal diversity in the intertidal sediments of China using the ITS2 rDNA metabarcoding approach [15], as well as other studies conducted in various marine habitats including coastal waters [19,20,26,41,42], the open sea [43], and even the deep biosphere [44,45], highlighting that these Dikarya members are ubiquitous in the ocean. Furthermore, our results also showed that the occurrence of these dominant lineages strongly shifted across habitats and seasons in the intertidal zones. For example, the members of Dothideomycetes prevailed in the sand beaches (i.e., FSB and SSB) and estuaries (i.e., EBS and EDG) in spring and winter, respectively, while the Eurotiomycetes and the Exobasidiomycetes were abundant in the estuaries during spring/summer and autumn, respectively. The basidiomycetous class Agaricomycetes was found to be the dominant fungal group in the anoxic-sulfidic mangrove sediments and is expected to degrade organic matter [9]. However, here, the Agaricomycetes abundantly occurred in the intertidal waters but not in the sediments. Moreover, this class was mainly represented by the genus *Amphinema*, which is known as the ectomycorrhizal fungi that commonly occur in soil [46]. Therefore, it is assumed that the abundant occurrence of *Amphinema* was probably related to the riverine inputs, as previously documented in coastal waters [18,20,26].

The abundant occurrence of six genera, i.e., *Cladosporium*, *Penicillium*, *Aspergillus*, *Alternaria*, *Fusarium*, and *Talaromyces* in the intertidal sediments, as also revealed in previous studies, highlights their ubiquitous nature and potential importance in the coastal ecosystem [15,19,26,42]. Some of them appeared to be resistant to polluted marine sediments and even have the ability to degrade marine pollutants. For example, some species affiliated with *Cladosporium*, (e.g., *C. herbarum*), *Alternaria* (e.g., *A. destruens*), or *Fusarium* (e.g., *F. pseudonygamai*) isolated from hydrocarbon-contaminated coastal sediments have been described for their tolerance and removal capacities towards polycyclic aromatic hydrocarbons in laboratory experiments [47,48,49]. Similarly, some marine representative isolates of these genera have recently been shown to be able to biodegrade some plastic polymers, such as *Cladosporium halotolerans* with polyurethane [50] and *Alternaria alternata* with polyethylene [51]. As it is known, intertidal zones are experiencing growing anthropogenic influences and suffering contamination. Therefore, it is hypothesized that the survival competitive advantage of fungi, derived from their adaptative abilities to cope with a wide range of pollutants, may explain their prevalence in intertidal sediments.

### 4.2. Potential Ecological Functions of Intertidal Fungi Switch with Habitat and Season

The diversity and composition of the fungal community can provide insights into their ecological roles with some degree of accuracy. The marine environment is commonly populated by genera such as *Alternaria*, *Fusarium*, *Epicoccum*, *Acremonium*, *Trichoderma*, and *Aureobasidium*, which serve as pathogens, symbionts, or decomposers of organic matter [52]. These findings show the intricate and varied ecological functions performed by fungi. The taxonomic makeup of fungal communities is closely linked to their roles in shaping the ecosystem they inhabit. This was supported by the FUNGuild analysis, where the above genera accounted for ~90% of the OTUs categorized in the multiple trophic mode of pathotroph–saprotroph–symbiotroph. Moreover, the occurrence of the fungi possessing this multiple trophic mode exhibited seasonal variation in different substrate types (sediment vs. seawater), as characterized by their prevalence in sediments and seawaters during winter and summer, respectively.

According to the FUNGuild analysis, the OTUs affiliated with *Talaromycetes*, *Emericellopsis*, *Phaeosphaeria*, *Nigrospora*, and *Tritirachium* were identified as saprotrophs. These genera mentioned above are, indeed, common saprophytic fungi and are often reported in salt marsh ecosystems where they play an important role in the degradation of organic detritus (plant leaves, residues, etc.) [6,53]. The two estuaries (i.e., EBS and EDG) investigated in this study are covered with a small amount of reed and seepweed plants, which can provide a suitable growth substrate for saprophytic fungi. Unexpectedly, the proportion of saprophytic fungi in the two estuaries (muddy beaches) is lower than that in the sandy beaches (i.e., FSB and SSB) in summer and autumn. The reason for this contradictory result is inferred here with caution. First, whether saprophytic fungi in an aquatic habitat can grow in large quantities, in addition to appropriate nutrient conditions, the level of oxygen is also very important, as highlighted by the fact that a low level of dissolved oxygen can severely inhibit fungal activity in terms of degradation of organic compounds in aquatic habitats [54]. Compared to the muddy sediments, the sandy sediments were found to have a higher level of daily oxygen flux [55], which might have been more conducive to the occurrence of the saprophytic process. Second, the two investigated swimming beaches directly face the Yellow Sea, which is often threatened by the outbreak of *Ulva* bloom during summer in recent years [56], which enriches intertidal sediments with organic matter as nutritional substrates for saprophytic fungi.

On the other hand, space–time variation in fungal ecological functions must be interpreted with caution because of the limitations of the FUNGuild tool in the prediction of trophic mode. The genera *Clasdiosporium*, *Aspergillus*, and *Penicillium* are common fungi in salt ecosystems worldwide; some of them are saprophytic fungi [6]. Meanwhile, the zoosporic fungi, with a small amount in our fungal dataset including chytrids and Rozellomycota, are known as pathogens of algae and other fungi [12,14,18,57]. These rare microbes would contribute more to ecosystem functioning than expected [58]. However, the trophic modes of the fungi mentioned above have not been recognized by the FUNGuild tool, which leads to the production of biases in the estimation of trophic mode. Despite this, such elementary results pave the way for a metatranscriptomic approach at the mRNA-expression level to delve deeper into the ecological role in such a dynamic setting.

### 4.3. Environmental and Anthropogenic Factors Strongly Influence the Intertidal Mycobiome

The microbial diversity and community composition in the ocean are strongly shaped by environmental selection [59]. Environmental variables such as temperature, salinity, pH, and nutrients were revealed to have significant influences on fungal communities in various marine ecosystems [18,19,20,26,60]. Here, our results showed that the variation in community structure in intertidal sediments could be well explained by temperature, salinity, pH, and TOC. In addition, a positive effect of temperature on the occurrence of saprotrophic fungi in sediments was found. This appears consistent knowing that many expressed genes involved in proteolysis, carbohydrate metabolic processes, and oxidoreductase activity are known to be temperature-sensitive in eukaryotes, including fungi [61,62]. This is also consistent with the idea that temperature is one of the critical factors that govern the distribution pattern of fungal diversity and community in the ocean [18,43,63,64].

Interestingly, we found a negative effect of TN on fungal richness in intertidal sediments, which is consistent with a previous study that reported a similar correlation of dissolved nutrients including NO_3_^−^ and PO_4_^3−^ with mycoplankton diversity in the coastal waters of the Yellow Sea [20]. This might be interpreted by the eutrophication of Chinese coastal waters due to increasing anthropogenic pollution since the 1980s [65,66], as characterized by the elevated nutrient concentrations in the water column that already had an inhibitory effect on biodiversity [67]. Similarly, more serious environmental pollution, such as microplastics and heavy metal contamination, has been detected in Qingdao’s bathing beaches in the peak tourist season, i.e., summer, compared to the other seasons [21,22], which might partially explain a relatively lower fungal richness in summer.

The factors including season, sampling location, and substrate type (sediment vs. seawater) essentially reflect a completive change of environmental physicochemical parameters and human activities in the space–time scale, which, together, inferred ~50% of the variation (R^2^ = 0.471, *p* = 0.001) in community structure in this study. This may highlight the complexity of environmental factors and the possible synergistic effects on microbial diversity that need further investigation.

## 5. Conclusions

This study revealed differentiated benthic and planktonic mycobiomes in the taxonomic composition and trophic mode that displayed seasonal and spatial dynamics in tidal flats, as characterized by the turnover of sediment communities with season and sampling location, whereas this turnover was experienced in seawater communities only with season. The trophic modes predicated also showed a seasonal pattern, as represented by abundant saprotrophic fungi in the summer, indicating a seasonality in the ecological roles performed by the fungal communities. The spatiotemporal patterns of fungal communities and their trophic modes were influenced by anthropogenic activities and weathers that jointly led to the changes in the physicochemical parameters of the seawaters and sediments in the intertidal zones of Qingdao, China. Our findings indicate the need for sound conservation and management strategies for maintaining these important ecosystems.

## Figures and Tables

**Figure 1 jof-09-01015-f001:**
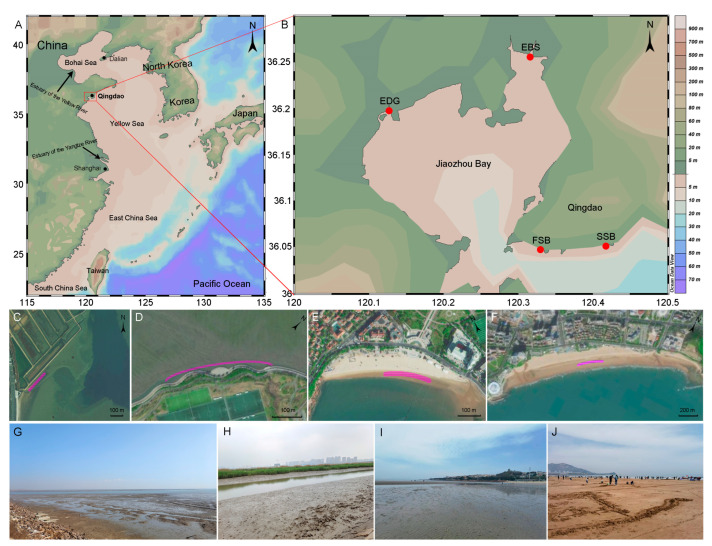
The four sampling locations (**A**,**B**) and the meso-tidal beaches sampled in EDG (**C**,**G**), EBS (**D**,**H**), FSB (**E**,**I**), and SSB (**F**,**J**). The pink curves in (**C**–**F**) mark the areas where sediment samples were collected; EDG and EBS are the estuaries of the Dagu and Baisha rivers, respectively; and FSB and SSB are the First and Shilaoren swimming beaches, respectively.

**Figure 2 jof-09-01015-f002:**
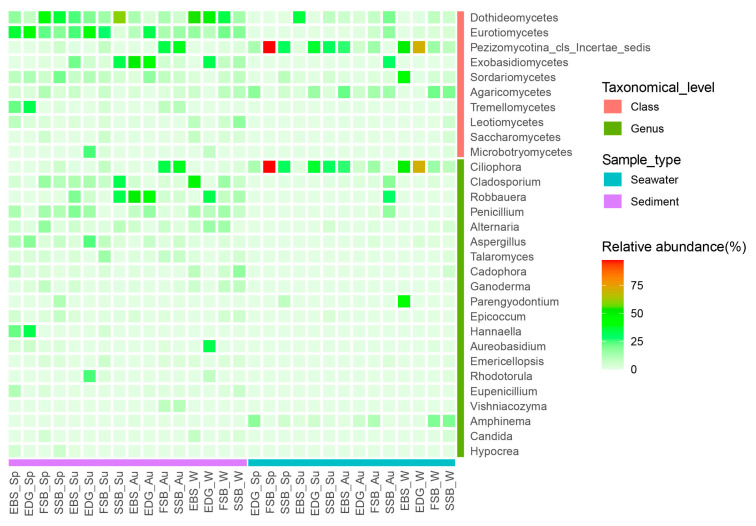
The relative read abundance of the top 20 genera and top 10 classes of the different sample groups separated by season and location (SW: seawater; SS: sediment; Sp: spring; Su: summer; Au: autumn; W: winter; EBS: the estuary of the Baisha River; EDG: the estuary of the Dagu River; and FSB and SSB indicate the First and Shilaoren swimming beaches, respectively).

**Figure 3 jof-09-01015-f003:**
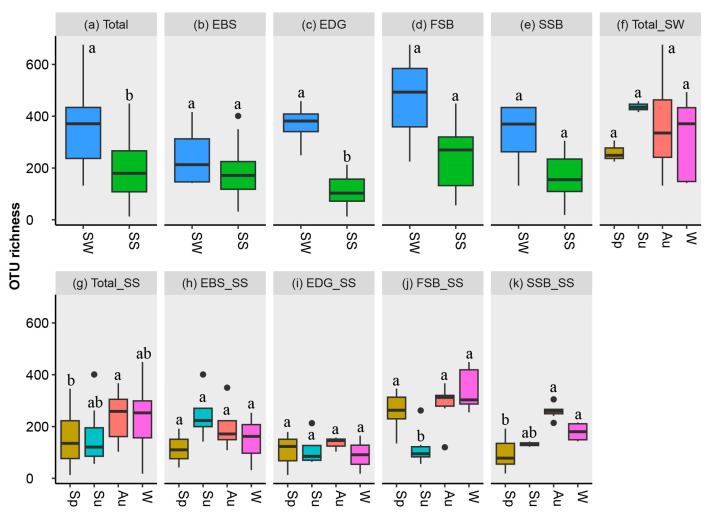
The OTU richness across the different substrates (seawater or sediment), seasons, and locations. SW: seawater; SS: sediment; Sp: spring; Su: summer; Au: autumn; and W: winter. EDG and EBS indicate the estuaries of the Dagu and Baisha rivers, respectively; FSB and SSB stand for the First and Shilaoren swimming beaches, respectively. The plots without shared letters indicate the significance at the level of a *p* value < 0.05.

**Figure 4 jof-09-01015-f004:**
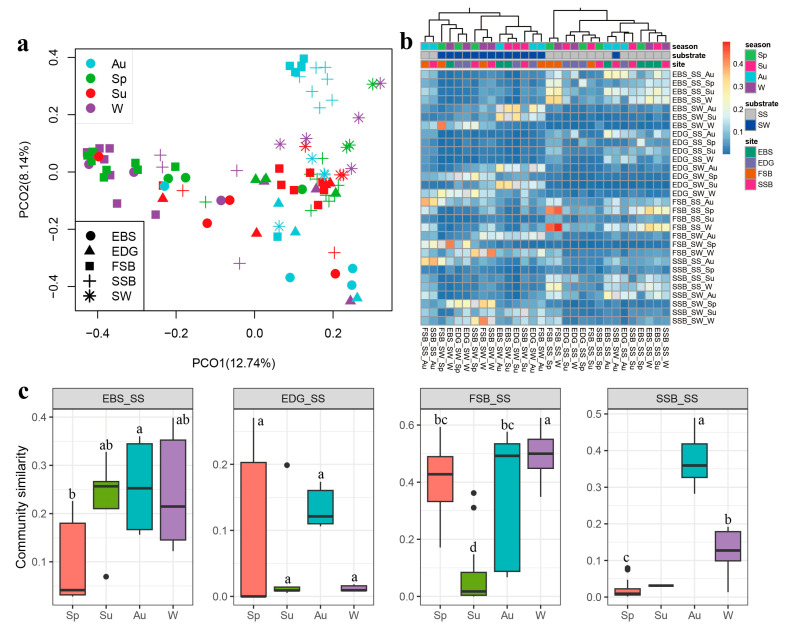
The principal coordinate analysis (PCoA) based on the Hellinger-transformed Bray–Curtis dissimilarity. (**a**) Comparison of community similarity (1 – Bray–Curtis index) among the different sampling groups (**b**) and seasons at the individual locations (**c**) SW: seawater; SS: sediment; Sp: spring; Su: summer; Au: autumn; and W: winter. EDG and EBS indicate the estuaries of the Dagu and Baisha rivers, respectively; FSB and SSB indicate the First and Shilaoren swimming beaches, respectively. The plots without shared letters indicate the significance at the level of a *p* value < 0.05. The unmarked groups indicate that no significant difference analysis was conducted. For SSB_SS (**c**), there were not enough summer samples available for a statistical analysis of community similarity.

**Figure 5 jof-09-01015-f005:**
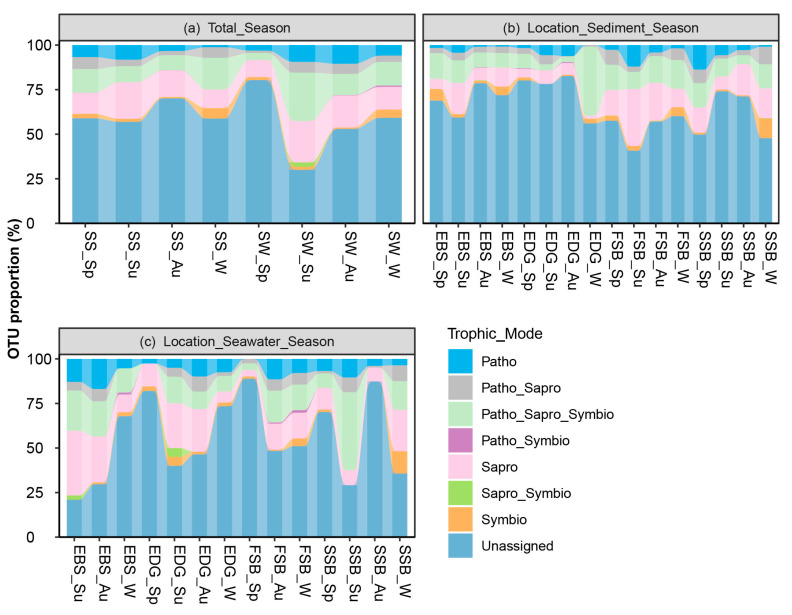
The OTU proportion of trophic modes in the different sampling groups separated by substrate and season (**a**) and by location, substrate (sediment (**b**), or seawater (**c**)) and season. Patho, Sapro, and Symbio are the abbreviations of Pathotroph, Saprotroph, and Symbiotroph, respectively. SW: seawater; SS: sediment; Sp: spring; Su: summer; Au: autumn; and W: winter. EDG and EBS indicate the estuaries of the Dagu and Baisha rivers, respectively; FSB and SSB indicate the First and Shilaoren swimming beaches, respectively.

## Data Availability

The fastq and barcode files were deposited in the National Center for Biotechnology Information Sequence Read Archive (SRA) as BioProject ID: PRJNA964474.

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
