# Peer review of "Seasonal and Spatial Dynamics of Fungal Diversity and Communities in the Intertidal Zones of Qingdao, China"

_jof, 2023, doi:10.3390/jof9101015_

Round 1

Reviewer 1 Report

The manuscript is of moderate importance as descriptive results and can be considered for publication in the journal, provided other reviewers agree for same. Some portions of the manuscript needs revisions.

1. Novelty statement needs to be rewritten. Currently it is quite confusing.

2. Figure 1 is important and informative but is too small to be descriptive. It can better be submitted as high resolution images as supplementary materials.

3. The conclusion is merely a mirror image of the abstract. It needs proper description of the whole results and a prospective hypothesis must be included that why there was variability in fungal presence in different locations and weathers.

4. Correct the minor typo and writing errors, such as in line 206-209.

none

Author Response

Reviewer: 1

Comments to the Author

The manuscript is of moderate importance as descriptive results and can be considered for publication in the journal, provided other reviewers agree for same. Some portions of the manuscript needs revisions.

R: We thank you for finding our results interesting. Please see our point by point responses.

  1. Novelty statement needs to be rewritten. Currently it is quite confusing.

R1: We rephased the part of Introduction, particularly the third and fourth paragraphs.

  1. Figure 1 is important and informative but is too small to be descriptive. It can better be submitted as high resolution images as supplementary materials.

R2: Thanks for your suggestion. A high resolution image of Figure 1 was provided in the main text.

  1. The conclusion is merely a mirror image of the abstract. It needs proper description of the whole results and a prospective hypothesis must be included that why there was variability in fungal presence in different locations and weathers.

R3: Thanks for your suggestion. We rephased this part.

  1. Correct the minor typo and writing errors, such as in line 206-209.

R4: Thanks. We carefully checked the typos and writing errors through the manuscript.

Reviewer 2 Report

This contribution aimed to explore spatiotemporal dynamics in fungal diversity, community composition, and potential functional roles and the impacts of environmental factors and human activities using an ITS2 metabarcoding approach in intertidal zones of China. Overall, this reviewer considers that the manuscript reports valuable, descriptive data on fungal communities. However, I detected major drawbacks that should be carefully considered as explained below:

I noticed major gaps in the introduction and relevant references are missing. Also, typos that should be carefully revised and corrected. All of the scientific names (genera and species) should be in italics. Please remember that the results are usually written in past tense, because they are describing the outcome of completed actions. So, check this and adjust the manuscript accordingly.

Ciliophora and Amphinema are not fungi. This is a fundamental error. Considering the title and aims of the investigation documenting this finding as part of the fungal community is misleading. Please revise. 

Minor comments:

Title: It is too long and confusing, also please check for grammar errors (use of articles). I suggest formulating a shorter and more concise title.

Abstract L18 and Results L235: Ciliophora and Amphinema are not fungi. This is a fundamental error. Considering the title and aims of the investigation documenting this finding as part of the fungal community is misleading. Please revise.

Abstract L23-24: This sentence is too vague for the abstract: “These shifts in fungal community composition could be explained partially by the environmental variables checked in this study including 24 temperature, salinity, and total organic carbon”. Please specify the trend.

Introduction L46-49: “Recently, a review listed 1,257 marine fungal 46 taxa recorded from a wide diversity of habitats (e.g., mangrove, salt marsh, deep-sea) and 47 substrates (e.g., drifting wood, sediment, plants, animals, algae) and occurring as sap- 48 robes, symbionts, parasites or mutualists [5]”. Please update the reference and numbers. As a suggestion, you could revise the latest volumes of the journal Botanica Marina, particularly:  Calabon, M. S., Jones, E. G., Pang, K. L., Abdel-Wahab, M. A., Jin, J., Devadatha, B., ... & Hyde, K. D. (2023). Updates on the classification and numbers of marine fungi. Botanica Marina66(4), 213-238.

Introduction L87-91: Please check the following argument, since numerous works have been published in this regard and should be properly acknowledged: “However, NGS-based studies associated with the spatiotemporal change in intertidal fungal communities are rare, except Picard [11] who observed an atypical seasonality of fungal communities in intertidal areas of North Carolina, as represented by more diverse fungal communities in cooler months. Thus, the extremely limited information hinders our understanding of the dynamics of intertidal mycobiome and the influence of environmental variables”.

Materials and Methods: relevant details on the sampling locations are missing. These include: country, latitude, longitude of sampling sites, dates, ecoregion, seasonality, etc.

Materials and Methods L112: Please cite a proper reference for this information “Both FSB and SSB are sand beaches that strongly suffer from human activities due to the massive influx of tourists during summer”.

Labels of Figure 1 are too small. Also, please consider that JoF aims to a international readership that is not necessarily familiar with the geography of China. So, a larger scale map would be useful to introduce your sampling sites. Please improve.

Bioinformatic Analysis: I am particularly concerned about the methodology selected to analyze Illumina data, considering the increasing popularity of denoising approaches (ASV-based methods), I would strongly recommend the authors to consider revising their approach and to implement ASVs instead of OTUs. As an example of this, please refer to: Chiarello, M., McCauley, M., Villéger, S., & Jackson, C. R. (2022). Ranking the biases: The choice of OTUs vs. ASVs in 16S rRNA amplicon data analysis has stronger effects on diversity measures than rarefaction and OTU identity threshold. PloS one, 17(2), e0264443.

Bioinformatic Analysis L180: check typo in “naïve Bayesian classifier”.

Bioinformatic Analysis L182: This reviewer is curious about the rationale for selecting a node support above 60%. In my experience this cutoff value is too low.

Bioinformatic Analysis L184: This reviewer understands the curiosity of the authors to infer functional from their dataset. However, FUNGuild tool was built/developed principally using data from terrestrial systems. In case of your data, only 3,527 OTUs (from 14,279 OTUs that is nearly 24% of the overall community) were used in this analysis, which is a major bias. Also, please consider the representation of members of the rare biosphere in your samples (Chytridiomycota, Mucoromycota, Rozellomycota, Mortierellomycota, and Aphelidiomycota) in the FUNGuild database. This is relevant as these rare taxa play a key role in ecosystem function. For more info please revise the classic paper: Jousset, A., Bienhold, C., Chatzinotas, A., Gallien, L., Gobet, A., Kurm, V., ... & Hol, W. H. (2017). Where less may be more: how the rare biosphere pulls ecosystems strings. The ISME journal, 11(4), 853-862. So, this reviewer considers that inferring any pattern from this sub-dataset may be misleading (the discussion needs to be redrafted in this sense, and special attention should be payed to the rare biosphere).

Bioinformatic Analysis L205: Please specify which environmental parameters?

Results L207-209: I suspect these lines were erroneously included “This section may be divided by subheadings. It should provide a concise and precise description of the experimental results, their interpretation, as well as the experimental conclusions that can be drawn”. 

The discussion falls short in the interpretation of the impacts of environmental factors and human activities in the fungal community, despite this was part of the objectives of the investigation.

Conclusions L440-442: This was not formally evaluated “A negative impact of anthropogenic pressure was observed at the FSB (swimming beaches) which exhibited lower 441 fungal diversity during peak tourism season (summer) compared to other seasons”.

Overall, I recommend a thorough English (for language and grammar) and style revision to avoid wordy, reiterative sentences (please check the correct use of articles).

Author Response

Reviewer: 2

Comments to the Author

This contribution aimed to explore spatiotemporal dynamics in fungal diversity, community composition, and potential functional roles and the impacts of environmental factors and human activities using an ITS2 metabarcoding approach in intertidal zones of China. Overall, this reviewer considers that the manuscript reports valuable, descriptive data on fungal communities. However, I detected major drawbacks that should be carefully considered as explained below:

R: We thank you for finding our results valuable. Please see our point by point responses.

I noticed major gaps in the introduction and relevant references are missing. Also, typos that should be carefully revised and corrected. All of the scientific names (genera and species) should be in italics. Please remember that the results are usually written in past tense, because they are describing the outcome of completed actions. So, check this and adjust the manuscript accordingly.

R1: Thanks for your suggestions. As you suggested, some reference articles were added at the right place. Typos were checked carefully through the main text and related files. About the issue that the scientific names should be italics, this was the mistake made by us during text conversion between common word version and the template provided by JoF. Sorry for this.

Ciliophora and Amphinema are not fungi. This is a fundamental error. Considering the title and aims of the investigation documenting this finding as part of the fungal community is misleading. Please revise. 

R2: Ciliophora and Amphinema are fungal genera. This can be checked online at Index Fungorum (http://www.indexfungorum.org/names/Names.asp) and MycoBank (https://www.mycobank.org/page/Basic%20names%20search).

Minor comments:

Title: It is too long and confusing, also please check for grammar errors (use of articles). I suggest formulating a shorter and more concise title.

R3: The previous title was replaced by new one as read “Seasonal and spatial dynamics of fungal diversity and communities in the intertidal zones of Qingdao, China”.

Abstract L18 and Results L235: Ciliophora and Amphinema are not fungi. This is a fundamental error. Considering the title and aims of the investigation documenting this finding as part of the fungal community is misleading. Please revise.

R4: As mentioned previously, the two genera are fungi. 

Abstract L23-24: This sentence is too vague for the abstract: “These shifts in fungal community composition could be explained partially by the environmental variables checked in this study including 24 temperature, salinity, and total organic carbon”. Please specify the trend.

 R5: We revised this sentence. Please check line 25-28.

Introduction L46-49: “Recently, a review listed 1,257 marine fungal 46 taxa recorded from a wide diversity of habitats (e.g., mangrove, salt marsh, deep-sea) and 47 substrates (e.g., drifting wood, sediment, plants, animals, algae) and occurring as sap- 48 robes, symbionts, parasites or mutualists [5]”. Please update the reference and numbers. As a suggestion, you could revise the latest volumes of the journal Botanica Marina, particularly:  Calabon, M. S., Jones, E. G., Pang, K. L., Abdel-Wahab, M. A., Jin, J., Devadatha, B., ... & Hyde, K. D. (2023). Updates on the classification and numbers of marine fungi. Botanica Marina, 66(4), 213-238.

R6: Thanks for your suggestions. We updated the information of marine fungal diversity and added the article as reference. Please check reference No.5.

Introduction L87-91: Please check the following argument, since numerous works have been published in this regard and should be properly acknowledged: “However, NGS-based studies associated with the spatiotemporal change in intertidal fungal communities are rare, except Picard [11] who observed an atypical seasonality of fungal communities in intertidal areas of North Carolina, as represented by more diverse fungal communities in cooler months. Thus, the extremely limited information hinders our understanding of the dynamics of intertidal mycobiome and the influence of environmental variables”.

 R7: We rephased this paragraph. Please check line 90-95.

Materials and Methods: relevant details on the sampling locations are missing. These include: country, latitude, longitude of sampling sites, dates, ecoregion, seasonality, etc.

R8: These details on the sampling locations can be found in Supplementary Table S1.

Materials and Methods L112: Please cite a proper reference for this information “Both FSB and SSB are sand beaches that strongly suffer from human activities due to the massive influx of tourists during summer”.

 R9: We added proper references. Please check line 138.

Labels of Figure 1 are too small. Also, please consider that JoF aims to a international readership that is not necessarily familiar with the geography of China. So, a larger scale map would be useful to introduce your sampling sites. Please improve.

 R10: Thanks for your suggestion. A high resolution image titled with Figure 1 was provided.

Bioinformatic Analysis: I am particularly concerned about the methodology selected to analyze Illumina data, considering the increasing popularity of denoising approaches (ASV-based methods), I would strongly recommend the authors to consider revising their approach and to implement ASVs instead of OTUs. As an example of this, please refer to: Chiarello, M., McCauley, M., Villéger, S., & Jackson, C. R. (2022). Ranking the biases: The choice of OTUs vs. ASVs in 16S rRNA amplicon data analysis has stronger effects on diversity measures than rarefaction and OTU identity threshold. PloS one, 17(2), e0264443.

R11: Thanks for the suggestions from the reviewer. ASVs and OTUs inference approaches are the two common methods that are used to cluster reads for further estimation of microbial diversity and community composition. DADA2 (ASVs inference method) supports analysis of distinct sequence variants, which may not be optimal for the fungal ITS region, given that it is sometimes present in multiple distinct copies per genome (Nilsson et al., 2019, Nature Reviews Microbiology, doi:10.1038/s41579-018-0116-y). We also looked through the reference Chiarello et al. (2022), and found that after rarefaction there was a similar trend of bacterial diversity among the OTUs- and ASVs-based methods. According to our experiences, the two methods could provide similar trend of fungal diversity that don’t influence the conclusions associated with fungal ecology.

Bioinformatic Analysis L180: check typo in “naïve Bayesian classifier”.

R12: We revised the correct writing as ‘Naïve Bayesian Classifier’.

Bioinformatic Analysis L182: This reviewer is curious about the rationale for selecting a node support above 60%. In my experience this cutoff value is too low.

R13: We agree the reviewer strongly. However, taking this cutoff value is a balance between more fungal taxa and accurate assignment.

Bioinformatic Analysis L184: This reviewer understands the curiosity of the authors to infer functional from their dataset. However, FUNGuild tool was built/developed principally using data from terrestrial systems. In case of your data, only 3,527 OTUs (from 14,279 OTUs that is nearly 24% of the overall community) were used in this analysis, which is a major bias. Also, please consider the representation of members of the rare biosphere in your samples (Chytridiomycota, Mucoromycota, Rozellomycota, Mortierellomycota, and Aphelidiomycota) in the FUNGuild database. This is relevant as these rare taxa play a key role in ecosystem function. For more info please revise the classic paper: Jousset, A., Bienhold, C., Chatzinotas, A., Gallien, L., Gobet, A., Kurm, V., ... & Hol, W. H. (2017). Where less may be more: how the rare biosphere pulls ecosystems strings. The ISME journal, 11(4), 853-862. So, this reviewer considers that inferring any pattern from this sub-dataset may be misleading (the discussion needs to be redrafted in this sense, and special attention should be payed to the rare biosphere).

R14: We agree the reviewer on this point. FUNGuild would produce a major bias on the prediction of ecological functions of fungal community. Therefore, we provided a caution about this issue as read at line 361-365. The paper mentioned by the reviewer was added at the line 459.

Bioinformatic Analysis L205: Please specify which environmental parameters?

R15: We revised as read ‘physicochemical parameters of seawater and sediment’ (line 259).

Results L207-209: I suspect these lines were erroneously included “This section may be divided by subheadings. It should provide a concise and precise description of the experimental results, their interpretation, as well as the experimental conclusions that can be drawn”. 

 R16: Sorry for this. These lines were deleted.

The discussion falls short in the interpretation of the impacts of environmental factors and human activities in the fungal community, despite this was part of the objectives of the investigation.

 R17: About the interpretation of the impacts of human activities can also be found from line 478-487.

Conclusions L440-442: This was not formally evaluated “A negative impact of anthropogenic pressure was observed at the FSB (swimming beaches) which exhibited lower 441 fungal diversity during peak tourism season (summer) compared to other seasons”.

 R18: We rephased the part of Conclusions.

Comments on the Quality of English Language. Overall, I recommend a thorough English (for language and grammar) and style revision to avoid wordy, reiterative sentences (please check the correct use of articles).

R19: As the reviewer suggested, we checked carefully through the main text and related files.

Round 2

Reviewer 2 Report

After reading the author's response, I am satisfied. However, prior to accepting the manuscript I would strongly recommend the authors to perform a small subsample of their data and run the analysis using a ASV-method just to confirm that the diversity among the OTUs- and ASVs-based methods is similar as they claim.

Author Response

Comments: After reading the author's response, I am satisfied. However, prior to accepting the manuscript I would strongly recommend the authors to perform a small subsample of their data and run the analysis using a ASV-method just to confirm that the diversity among the OTUs- and ASVs-based methods is similar as they claim.

Response: Thanks for your positive comments. Yes, we did an analysis to confirm our claim by running ASV- and OTU-based methods using a small subsample of our data. Similar results were obtained between the above two methods.